# Ecoregional and Phytogeographical Insights into the Distribution of *Tulipa* in the 'Nature Imperiled' Area of Central Asia for Effective Conservation

Temur Asatulloev [1,2,3,4,5], Davron Dekhkonov [3], Ziyoviddin Yusupov [2,6], Umida Tojiboeva [5], Lei Cai [1], Komiljon Tojibaev [6,*] and Weibang Sun [1,4,*]

1   Yunnan Key Laboratory for Integrative Conservation of Plant Species with Extremely Small Populations, Kunming Institute of Botany, Chinese Academy of Sciences, Kunming 650201, China; asatullayevtemurrr@gmail.com (T.A.); cailei@mail.kib.ac.cn (L.C.)
2   Yunnan International Joint Laboratory for Biodiversity of Central Asia, Kunming Institute of Botany, Chinese Academy of Sciences, Kunming 650201, China; yusupov@mail.kib.ac.cn
3   Department of Biology, Namangan State University, Namangan 716019, Uzbekistan; davron-b@bk.ru
4   CAS Key Laboratory for Plant Diversity and Biogeography of East Asia, Kunming Institute of Botany, Chinese Academy of Sciences, Kunming 650201, China
5   University of Chinese Academy of Sciences, Beijing 100864, China; tojiboeva.umida@mail.ru
6   Institute of Botany, Academy of Sciences, Tashkent 100125, Uzbekistan
*   Correspondence: ktojibaev@mail.ru (K.T.); wbsun@mail.kib.ac.cn (W.S.)

**Abstract:** *Tulipa* L. (Liliaceae) comprises approximately 150 species. Although Central Asia, the main center of its diversity, includes around 66 species, detailed mapping of their distribution is limited and research on their ecoregional and phytogeographical dispersion is insufficient. This study aimed to map and analyze the distribution patterns of *Tulipa* across the Central Asian ecoregions and phytogeographical regions to identify potential hotspots for effective conservation efforts. The results identified the Gissaro-Alai open woodlands ecoregion, which hosts 41 species of *Tulipa*, as the leading hotspot ecoregion. The Ferghana Valley phytogeographical district (Afghano-Turkestan province) was found to be the most suitable habitat for 25 species of *Tulipa*. We also determined that altitude has a strong influence on the diversity of *Tulipa* and indicating increase of species richness as elevation rises. However, as elevation rises up from ca. 2000 m a.s.l. species richness also decreases slightly. An analysis of the distribution of sections of *Tulipa* in Central Asia revealed that around 61% of all species dwell in this area and that the sections *Kolpakowskianae* and *Biflores*, which have their greatest diversity of species, are specific to this area. The findings provide valuable insights into the distribution of *Tulipa* and allow for feasible recommendations and suggestions for their conservation.

**Keywords:** Central Asia; conservation; ecoregion; phytogeographic districts; *Tulipa*; distribution; phytogeography





## 1. Introduction

### 1.1. Central Asia as a "Nature Imperiled" Region

Central Asia is a vast region that encompasses Kazakhstan, Uzbekistan, Turkmenistan, Kyrgyzstan and Tajikistan. This region is known for the diversity and uniqueness of its plant species, which have adapted to the harsh climatic conditions of the area. However, Central Asia is also considered to be a "nature imperiled" area due to human activities, such as deforestation, overgrazing and climate change, all of which have led to a decline in plant diversity in the region [1,2]. According to Khassanov [3], Central Asia has a rich flora of almost 10,000 species of vascular plants, including many endemic species. Many of these species, however, are threatened by habitat loss and degradation, as well as by overexploitation for medicinal and other purposes. The International Union for the Conservation of Nature (IUCN) Red List of Threatened Species [4] reports that several plant

species in Central Asia are critically endangered, endangered, or vulnerable. The decline in the diversity of plant species in Central Asia is part of a larger global trend of biodiversity loss in the Anthropocene era [5]. This loss of biodiversity has significant ecological and socio-economic consequences, including the disruption of ecosystem services and the loss of cultural and traditional knowledge associated with plants.

### 1.2. Species of Tulipa as Part of Central Asian Plant Diversity

*Tulipa* L. (Liliaceae), a monocot genus primarily in Central Asia, extends into southeastern Europe, the Middle East and across North Africa [6,7]. Due to its horticultural value and cultural significance, it has garnered significant attention [7]. The four subgenera, *Orythia, Clusianae, Eriostemones* and *Tulipa*, have been further subdivided into 10 sections [6]. Central Asia is considered to be the hotspot center for *Tulipa* [8]. Four subareas where tulips grow were identified by Botschantzeva [8]: the deserts and semi-deserts of the Turan plains, the Pamir-Alay mountains, the Western Tian Shan mountains and the Central Asian highlands. The Tian Shan and Pamir-Alay mountains of Central Asia are the primary centers of diversity for *Tulipa* [8]. Field surveys and taxonomic and related studies of *Tulipa* have been carried out by Tojibaev and Beshko [9], Lazkov and Pashinina [10], Valdschmit [11], Wilson, Dolotbakov et al. [12], Dekhkonov, Tojibaev et al. [13], Asatulloev, Dekhkonov et al. [14], Dekhkonov, Asatulloev et al. [15], Tojibaev, and Dekhkonov et al. [16] across Central Asia. Their distribution, IUCN categories and conservation measures were investigated and developed for tulips at the country level for Uzbekistan as well [17]. Currently, 50 species of *Tulipa* in the flora of Central Asia appear on the IUCN's red list as threatened and 36 species are included in the national Red Data Books of the Central Asian countries. Additionally, 39 species were reported to be in natural protected areas in various Central Asian countries.

### 1.3. Ecoregions as Determinants of Plant Diversity

Ecoregions are areas defined by distinct ecological characteristics, such as climate, geology and vegetation [18]. They play a crucial role in determining the distribution of plant species, as different plant species have adapted to the specific environmental conditions of each of the different ecoregions [19]. An ecoregion characterized by a unique climate may lead to the evolution of plant species adapted to drought, hot and other environmentally unique conditions resulting in a high level of endemism [20]. The concept of "ecoregions" serves various important purposes. It assists in evaluating contrasting environmental values, determining priorities for allocating limited resources, and facilitating effective actions, such as acquiring protected areas and restoring habitats. Understanding the distribution of plant species across ecoregions is important for conservation efforts, as it can help to identify areas with high levels of biodiversity and prioritize conservation efforts in those areas. For this purpose, the identification of biodiversity hotspots has typically relied on species richness and many studies have been conducted based on this idea [21–24]. Thus, counting species richness, uniqueness and abundance within an ecoregion provides insight into the role of that ecoregion in shaping biodiversity and its importance. In addition, as a part of ecoregions, elevation and environmental gradients play significant roles within an ecoregion, affecting the diversity and distribution of species [25]. They provide valuable opportunities for studying the effects of environmental factors, such as temperature, on the ecological and evolutionary responses of living organisms [26] The gradients are considered to be natural experiments with significant potential for research. In support of this idea, in Uzbekistan, elevation was determined to be one of the most important factors, with an elevational range of 700–2200 m being the most suitable for species of *Tulipa* [14].

### 1.4. Role of Phytogeographical Analysis on Plant Diversity

Phytogeographical distribution analysis is a crucial tool for understanding the distribution of plant species and their habitats. It is a subfield of biogeography that focuses on the spatial distribution of plant species and the factors that influence their distribution [27]. In addition, it provides insights into the natural distribution patterns of plants, including

their historical ranges and the environmental conditions they require for survival. By analyzing the distribution of plant species across different regions, we can identify areas of high biodiversity and prioritize conservation efforts in those areas. For example, *Tulipa* species in the natural geographic area of the Ferghana Valley (phytogeographic district) were studied and the importance of this valley for tulips was provided [28]. A case study about floristic diversity based on species richness of Tarvagatai Nuruu National Park was carried out and found special habitat types that must be prioritized for conservation efforts [29]. In those studies, researchers mostly emphasized endemism, species richness and vegetation composition to determine most priority areas for conservation. As species richness, endemism and vegetation composition are most of the important components of phytogeographical delineation, identifying these characterizations of phytogeographic districts may provide crucial feedback on conservation and we tried this for a genus level (*Tulipa* L.). In this, we follow the regionalization proposed by Kamelin [27], since it is the most recently published one. According to Kamelin, Soviet Middle Asia covers areas from the east to Xinjiang, from the west to the Caspian Sea, from the north to the southern part of Kazakhstan (not including the northern part), from the northeast to Mongolia (a certain part of Mongolia), and from the south to the northern part of Afghanistan. Kamelin divided the Soviet Middle Asia flora into 8 provinces and 78 districts.

Thus, our aims from mapping were:

1. To identify potential hotspots for the conservation of species of *Tulipa*.
2. To determine elevational pattern of tulips distribution throughout Central Asia.
3. To fill in gaps in the ecoregional and phytogeographical dispersion of species of *Tulipa* in Central Asia.
4. To provide valuable insights for conserving the species of *Tulipa*.
5. To make conservation recommendations or suggestions based on our findings.

## 2. Materials and Methods

We collected 1300 herbarium specimens of 66 species of *Tulipa* from mid-March to the end of July from 2014 to 2023 at altitudes ranging from 50 to 3650 m. Some areas were examined several times to ensure proper coverage and we recorded geographical coordinates. The 1300 specimens were deposited in the National Herbarium of Uzbekistan (TASH). The other sources of data used in this study were from previously published sources [8,30,31], herbarium sheets stored in TASH (National Herbarium of Uzbekistan), LE (Herbarium of Vascular Plants of the Komarov Botanical Institute) and MW (Moscow University Herbarium) [32], as well as information from the Global Biodiversity Information Facility (GBIF) [33] and Plantarium [34]. The species were identified according to Veldkamp and Zonneveld, Tojibaev, and Dekhkonov et al. [6,16] and cross-checked in Plants of the World Online [35]. Overall, 66 species of *Tulipa* were recognized in this study.

The study was based on the distribution of *Tulipa* in the ecoregions of Central Asia and botanical-geographical districts of Soviet Middle Asia. The terrestrial ecoregions of Central Asia were cropped from a terrestrial world ecoregion map [2] in ArcGIS v10.8. The phytogeographical map of Soviet Middle Asia created by Kamelin [27] was drawn manually and georeferenced in ArcGIS v10.8 [36]. We then enumerated species richness (66 species for ecoregions and 65 species for phytogeographical districts) in both maps. For phytogeographical analysis, we excluded *T. turgaica* as it was off the map. We then calculated how many species each ecoregion and phytogeographic district holds. After calculation, we visualized maps with species richness coloring by species number in ArcGIS v10.8 [36]. In addition, we visualized sectional distribution of 66 species within Central Asia using ArcGIS v10.8. Subsequently, we visualized barplots to indicate ecoregions and phytogeographic districts species richness and visualized interactive plots to see a general view of species and ecoregions/phytogeographic districts in which they are distributed. Additionally, we grouped *Tulipa* species according to occurrence in ecoregions/phytogeographic districts in order to determine the dispersal range. Elevation ranges were assessed based on geographic coordinates using GPS visualizer (https://www.gpsvisualizer.com, accessed

on 12 August 2022). After calculating elevational range for each species, we divided the upper (4450 m a.s.l.) and lower (50 m a.s.l.) range into 50 m bands. We then calculated how many species each band holds. After that, we determined and visualized the relationship between elevational range and species richness (for 66 species). Except for species richness and sectional distribution maps, all other visualizations were conducted in R [37].

## 3. Results

### 3.1. Ecoregional Analysis

In our study, we created a map of the species richness of *Tulipa* based on ecoregions (Figure 1). Our surveys revealed that the Pamir-Alay and Western Tian Shan were hotspots for tulips, with Gissaro-Alai open woodlands (GAOW) identified as the leading hotspot, hosting approximately 41 species (Figure 2A). The Alai-Western Tian Shan steppe (AWTSS) and Tian Shan foothills arid steppe (TSFAS) ecoregions followed, with 18 and 19 species, respectively, each comprising half of the total of the GAOW's species richness. The Tian Shan montane steppe and meadows (13 species, TSSM), Central Asian northern desert (10 species, CAND) and Central Asian southern desert (9 species, CASD) had moderate species richness. The Pamir alpine desert and tundra (7 species, PADT), Badghyz and Karabil semi-desert (6 species, BKSD) and Kazakh semi-desert (6 species, KSD) ecoregions had slightly greater species richness compared to the Tian Shan montane conifer forests (5 species, TSMCF) and Central Asian riparian woodlands (4 species, CARW), Altai alpine meadow and tundra (3 species, AAMT), Altai steppe and semi-desert (3 species, ASSD), Emin Valley steppe (3 species, EVS) and Jungar Basin semi-desert (3 species each), Kopet Dag woodlands and forest steppe (2 species, KDWFS), Kazakh steppe (2 species, KS), Caspian lowland desert (2 species, CLD), and Altai montane forest and forest steppe (1 species, AMFFS).

The species were categorized based on the number of ecoregions they inhabit, as demonstrated in Table 1 and Figure 2B and the interaction (or relatedness) between species and ecoregions in Figure 2C. The findings indicate that 2 species each from group 1 occur in the six same or different ecoregions, while 6 species each from group 2 occupy the five same or different ecoregions. Additionally, 7 species each from group 3 are present in the four same or different ecoregions, 9 species each from group 4 inhabit the three same or different ecoregions, 18 species each from group 5 are in the two same or different ecoregions, and 25 species each from group 6 are in the single same or different ecoregions. The habitat of each species in each ecoregion is provided in the supplementary file, Table S1.

Our analysis revealed an altitude and species richness relationship, with a clear peak in species diversity between 1000 and 2200 m above sea level (Figure 3). At this elevation range, we observed more than 35 different species, indicating a strong association between altitude and species richness in the study area.

### 3.2. Phytogeographical Analysis

The species richness map of species of *Tulipa* is based on the botanical-geographical regionalization of Middle Asia (Figure 4) and the species richness graph created for phytogeographic districts (Figure 5A). The botanical-geographical regionalization map shows the Ferghana valley, containing 25 species, to be the most diverse hotspot for tulips followed by Mogoltau-Kurama (13 species), Kukhistan (10 species), Gissar, Darvaz, Kugitang-Baisun (9 species each), Myunkum, Nuratau and Northern Tian Shan (8 species each), Kyzylkum, Chu-Ili and Western Gissar (7 species each), Ferghana (from Southern Turan province) and Balkhash surround (6 species each), Aral, Fergano-Alai and Northern Jungar (5 species each), Irtish-Urunguy, Southern Jungar (from mountainous Jungar province), Sonkul, Chernoirtish and Kopetdag-khorassan province (4 species each), Betpakdala, Pandj, Ketten, Terskey-Tengri and Ayaguz (3 species each), Lower Amudarya, Bukhara, Kashgar province, Kuldjin, Agadyr, Narym, Prizaysan, Southern Altai and Khankhuey (2 species each), Southern Jungar (from Jungar Province), Badakhshan, Central Gindukush, Alai, Karkaral and

Western Alai (1 species each). The remaining districts or provinces had no species of *Tulipa* or were data-deficient.

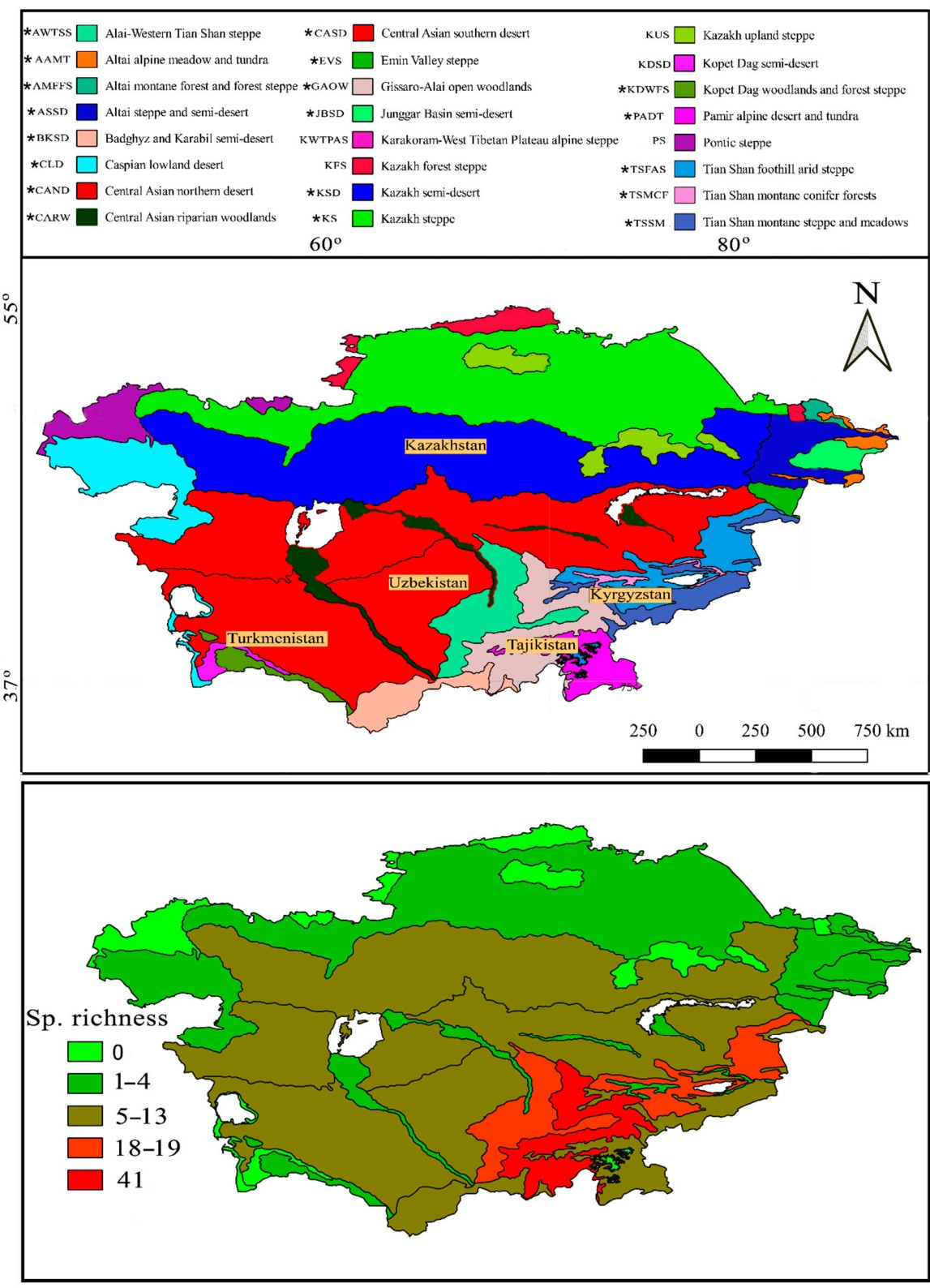

**Figure 1.** Ecoregions of Central Asia (above) [2], and ecoregions with asterisks hold at least one species. Species richness map of species of *Tulipa* (below) in these ecoregions (additionally, Table S1 provides information about each ecoregion and its corresponding species).

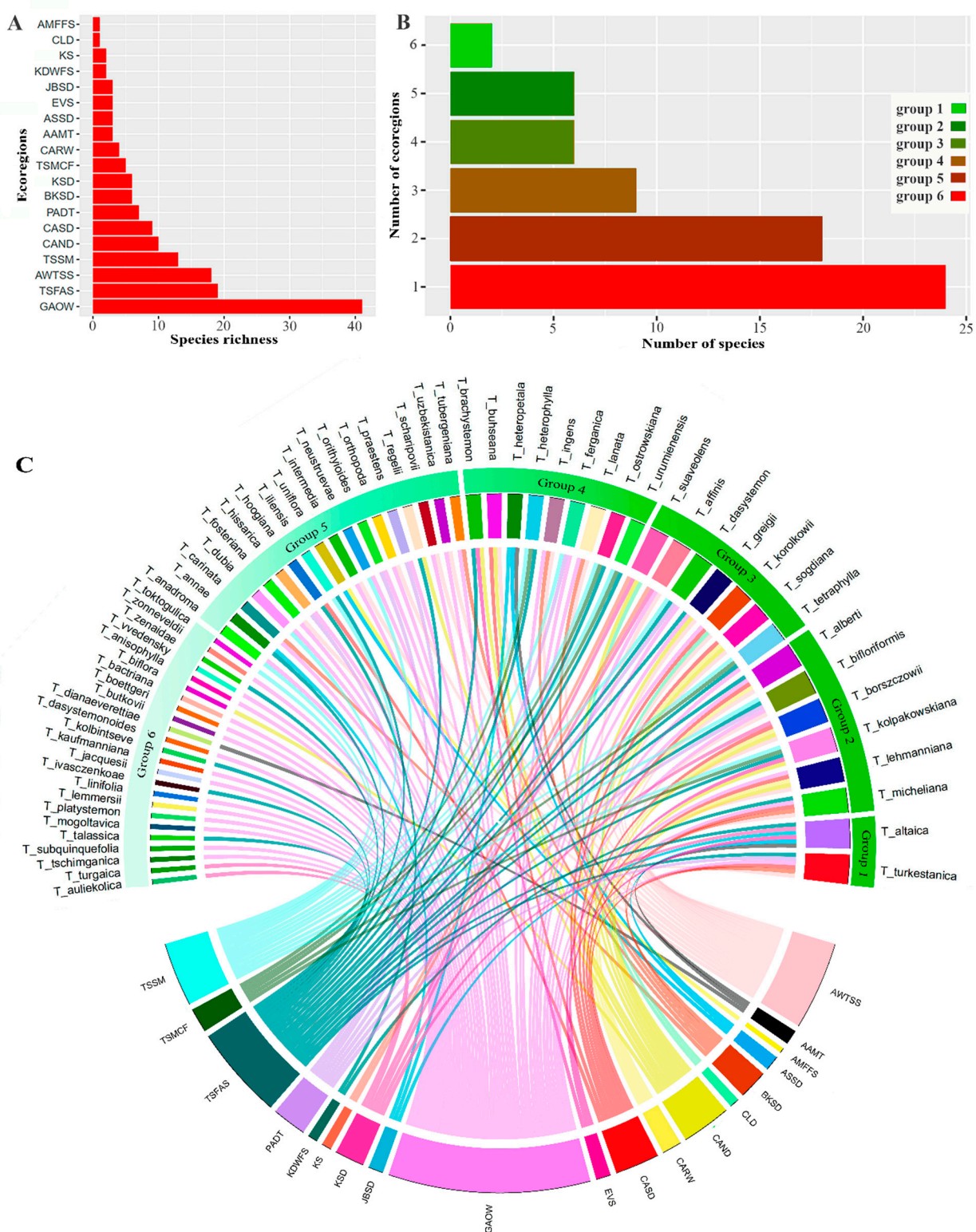

**Figure 2.** Species richness of ecoregions (**A**) and number of species, each distributed in how many ecoregions (**B**) and general view of species and ecoregions in which they are distributed (**C**).

**Table 1.** Groups of tulips according to the number of ecoregions they occupy.

| Groups | Species | Number of Phytogeographic Regions |
|---|---|---|
| **1** | *T. altaica, T. turkestanica* | 6 |
| **2** | *T. alberti, T. bifloriformis, T. borszczowii, T. kolpakowskiana, T. lehmanniana, T. micheliana* | 5 |
| **3** | *T. affinis, T. dasystemon, T. greigii, T. korolkowii, T. sogdiana, T. tetraphylla, T. suaveolens* | 4 |
| **4** | *T. brachystemon, T. buhseana, T. ferganica, T. heteropetala, T. heterophylla, T. ingens, T. lanata, T. ostrowskiana, T. urumiensis* | 3 |
| **5** | *T. anadroma, T. annae, T. carinata, T. dubia, T. fosteriana, T. hissarica, T. hoogiana, T. iliensis, T. intermedia, T. neustruevae, T. orithyioides, T. orthopoda, T. praestens, T. regelii, T. scharipovii, T. tubergeniana, T. uniflora, T. uzbekistanica* | 2 |
| **6** | *T. anisophylla, T. bactriana, T. biflora, T. boettgeri, T. butkovii, T. dasystemonoides, T. dianaeverettiae, T. ivasczenkoae, T. jacquesii, T. kaufmanniana, T. kolbintsevii, T. lemmersii, T. linifolia, T. mogoltavica, T. platystemon, T. subquinquefolia, T. talassica, T. tschimganica, T. toktogulica, T. turgaica, T. vvedensky, T. zenaidae, T. zonneveldii, T. auliekolica* | 1 |

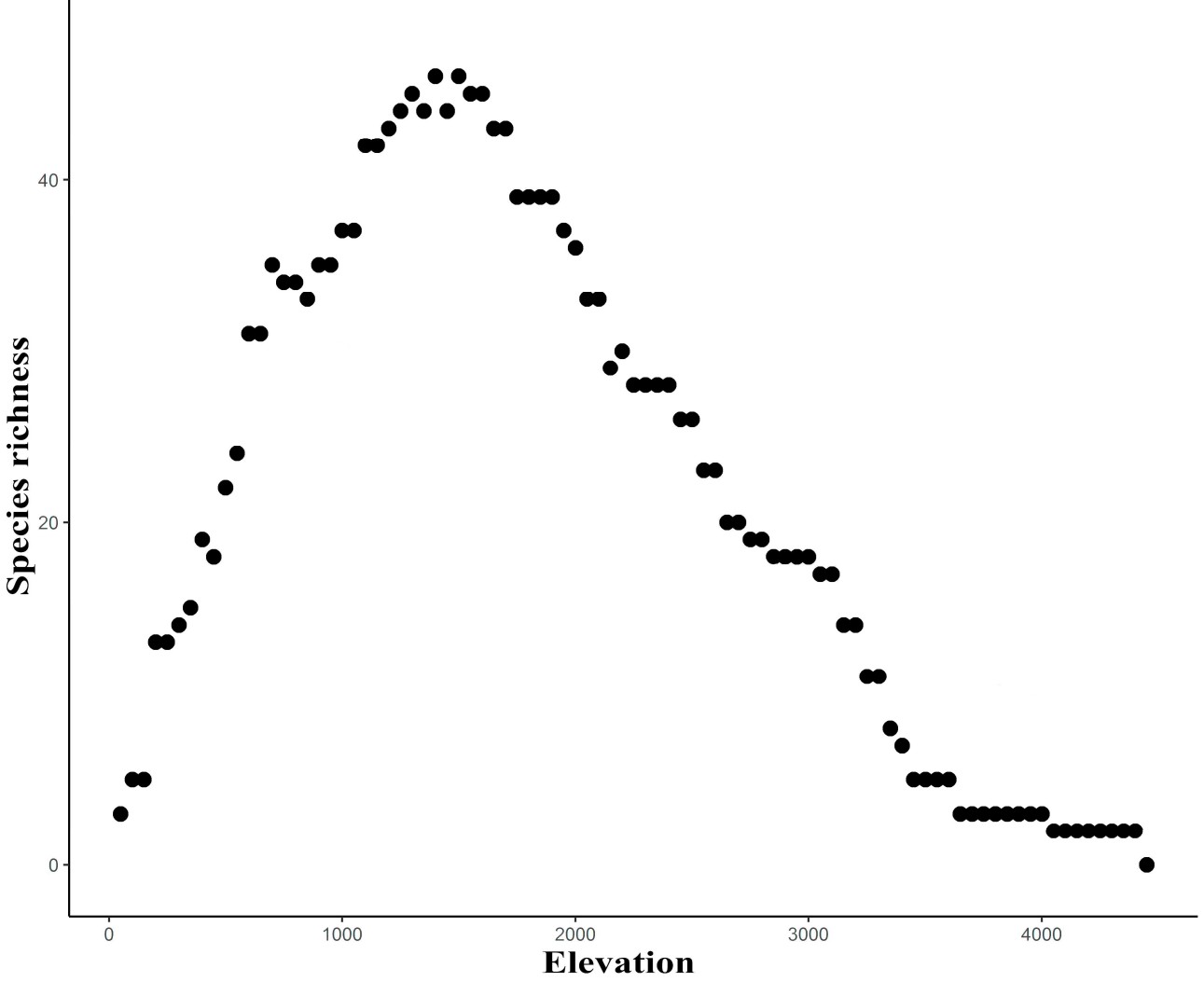

**Figure 3.** Relationship between richness and elevation for species of *Tulipa*.

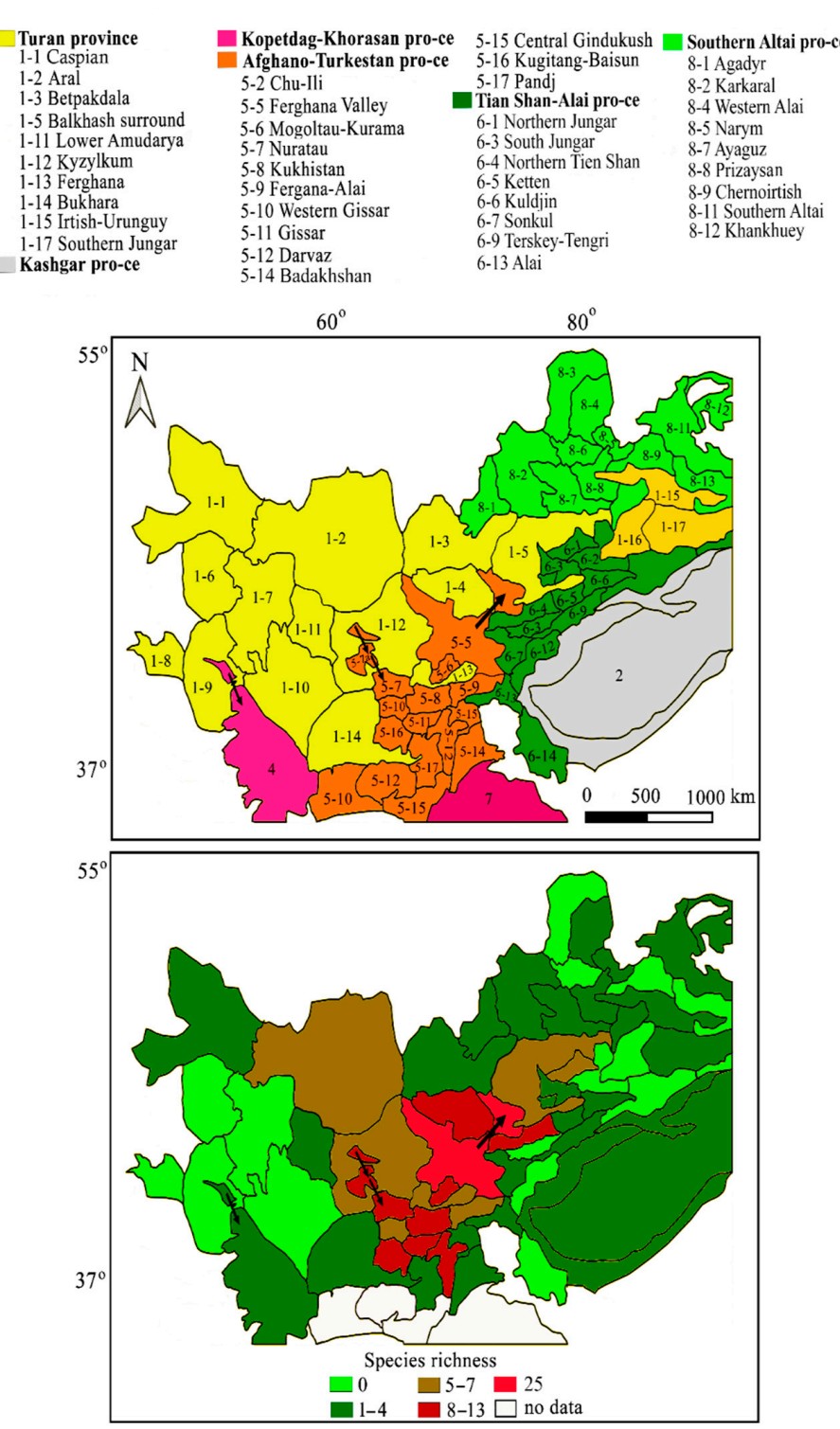

**Figure 4.** Botanical-geographical regionalization of Soviet Middle Asia [23] above and species richness map of species of *Tulipa* (below) in these regions. Note that only districts or provinces with species of *Tulipa* are given (Table S2 additionally provides information about each district/province and its corresponding species).

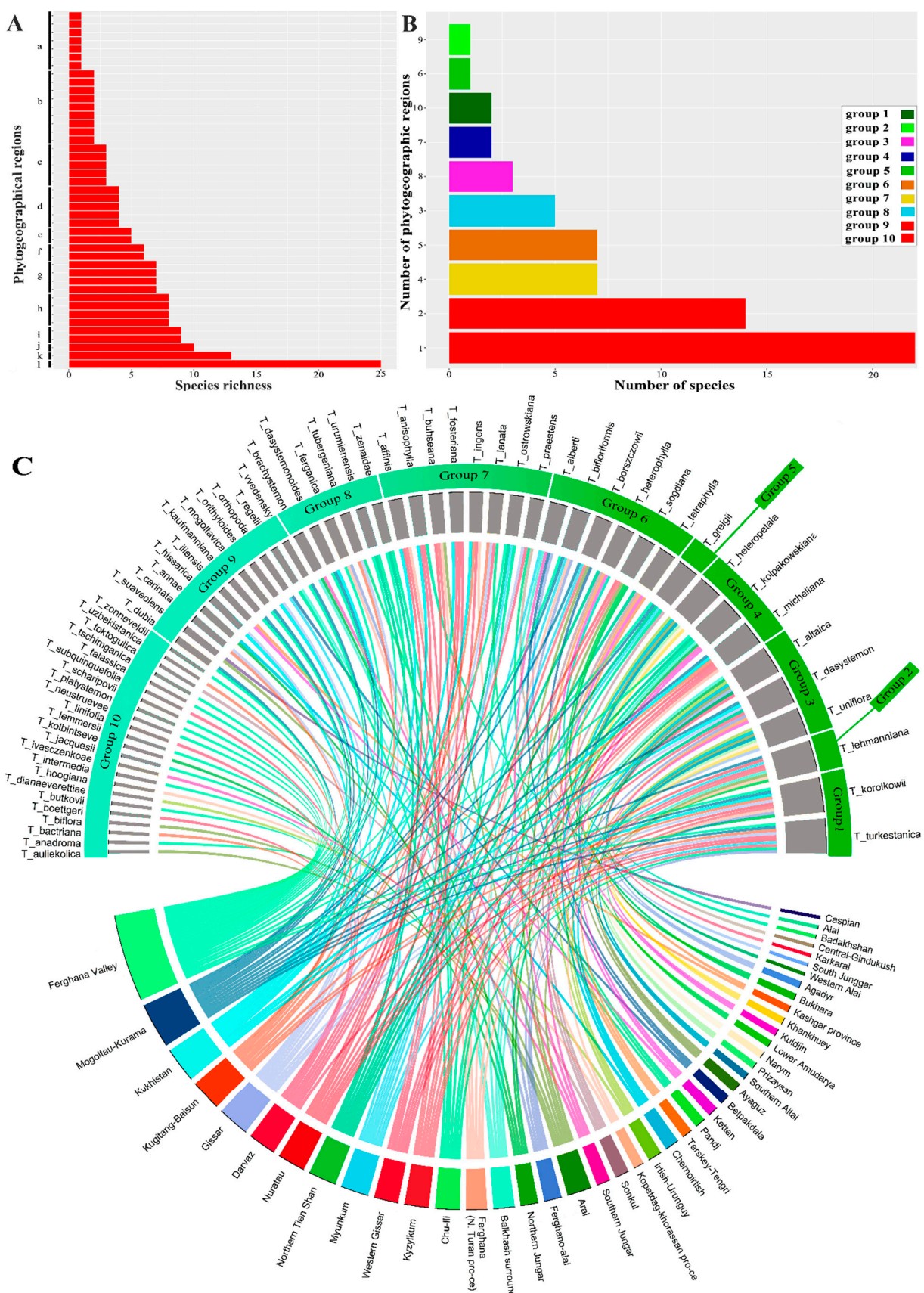

**Figure 5.** Species richness of phytogeographical regions (**A**): a-Caspian, Alai, Central Gindukush, Karakal, South Junggar (Junggar province), Western Alai; b-Agadyr, Bukhara, Kashgar province,

Khankhuey, Kuldjin, Lower Amudarya, Narym, Prizaysan, Southern Altai; c-Ayaguz, Betpakdala, Ketten, Pandj, Terskey-Tengri; d-Chernoirtish, Irtish-Urunguy, Kopetdag-khorassan province, Sonkul, South Junggar (Mt. Junggar province), e-Ferghano-alai, Northern Jungar; f-Balkhash surround, Ferghana (Northern Turan province); g-Chu-Ili, Aral, Kyzylkum, Western Gissar; h-Myunkum, Northern Tien Shan, Nuratau, Darvaz; i-Gissar, Kugitang-Baisun; j-Kukhistan; k-Mogoltau-Kurama; l-Ferghana Valley (**B**) number of species distributed in how many ecoregions (**C**) general view of species and ecoregions in which they are distributed.

The species were classified based on the number of phytogeographic districts they inhabit, as presented in Table 2 and Figure 5B and the interaction (or relatedness) between species and phytogeographic districts in Figure 5C. The results indicate that two species from group 1 occupy 10 phytogeographic districts, while one species from group 2 exists in 9 phytogeographic districts. Furthermore, three species each from group 3 are present in 8 of the same or different phytogeographic districts, three species each from group 4 inhabit 7 of the same or different phytogeographic districts, and one species from group 5 is in 6 phytogeographic districts. Additionally, six species each from group 6 are in 5 of the same or different phytogeographic districts, eight species each from group 7 are in 4 of the same or different phytogeographic districts, six species each from group 8 inhabit 3 of the same or different phytogeographic districts, 12 species each from group 9 are in 2 of the same or different phytogeographic districts, and 23 species each from group 10 inhabit a single phytogeographic district, either the same or different.

**Table 2.** Groups of tulips according to the number of phytogeographic regions they occupy.

| Groups | Species | Number of Phytogeographic Regions |
|---|---|---|
| 1 | *T. korolkowii, T. turkestanica* | 10 |
| 2 | *T. lehmanniana* | 9 |
| 3 | *T. altaica, T. dasystemon, T. uniflora* | 8 |
| 4 | *T. heteropetala, T. micheliana, T. kolpakowskiana* | 7 |
| 5 | *T. greigii* | 6 |
| 6 | *T. alberti, T. bifloriformis, T. borszczowii, T. heterophylla, T. sogdiana, T. tetraphylla* | 5 |
| 7 | *T. affinis, T. anisophylla, T. buhseana, T. fosteriana, T. ingens, T. lanata, T. praestens, T. ostrowskiana* | 4 |
| 8 | *T. brachystemon, T. dasystemonoides, T. ferganica, T. tubergeniana, T. urumiensis, T. zenaidae* | 3 |
| 9 | *T. annae, T. carinata, T. dubia, T. hissarica, T. iliensis, T. kaufmanniana, T. mogoltavica, T. orithyioides, T. orthopoda, T. regelii, T. vvedensky, T. suaveolens* | 2 |
| 10 | *T. anadroma, T. bactriana, T. biflora, T. boettgeri, T. butkovii, T. dianaeverettiae, T. hoogiana, T. intermedia, T. ivasczenkoae, T. jacquesii, T. kolbintsevii, T. lemmersii, T. linifolia, T. neustruevae, T. platystemon, T. scharipovii, T. subquinquefolia, T. talassica, T. tschimganica, T. toktogulica, T. uzbekistanica, T. zonneveldii, T. auliekolica* | 1 |

### 3.3. Sectional Distribution

Overall, 66 species from 10 sections (four subgenera) were recognized and mapped within Central Asia (Figure 6). Subgenus Orythia sect. Orythia included three species. Subgenus Tulipa, a large group comprising six sections: sect. Kolpakowskianae had 22 species, sect. Lanatae had 10 species, sect. Vinistriatae had 7 species, sect. Spiranthera had 4 species, sect. Tulipanum had 2 species and sect. Multiflorae had 1 species.

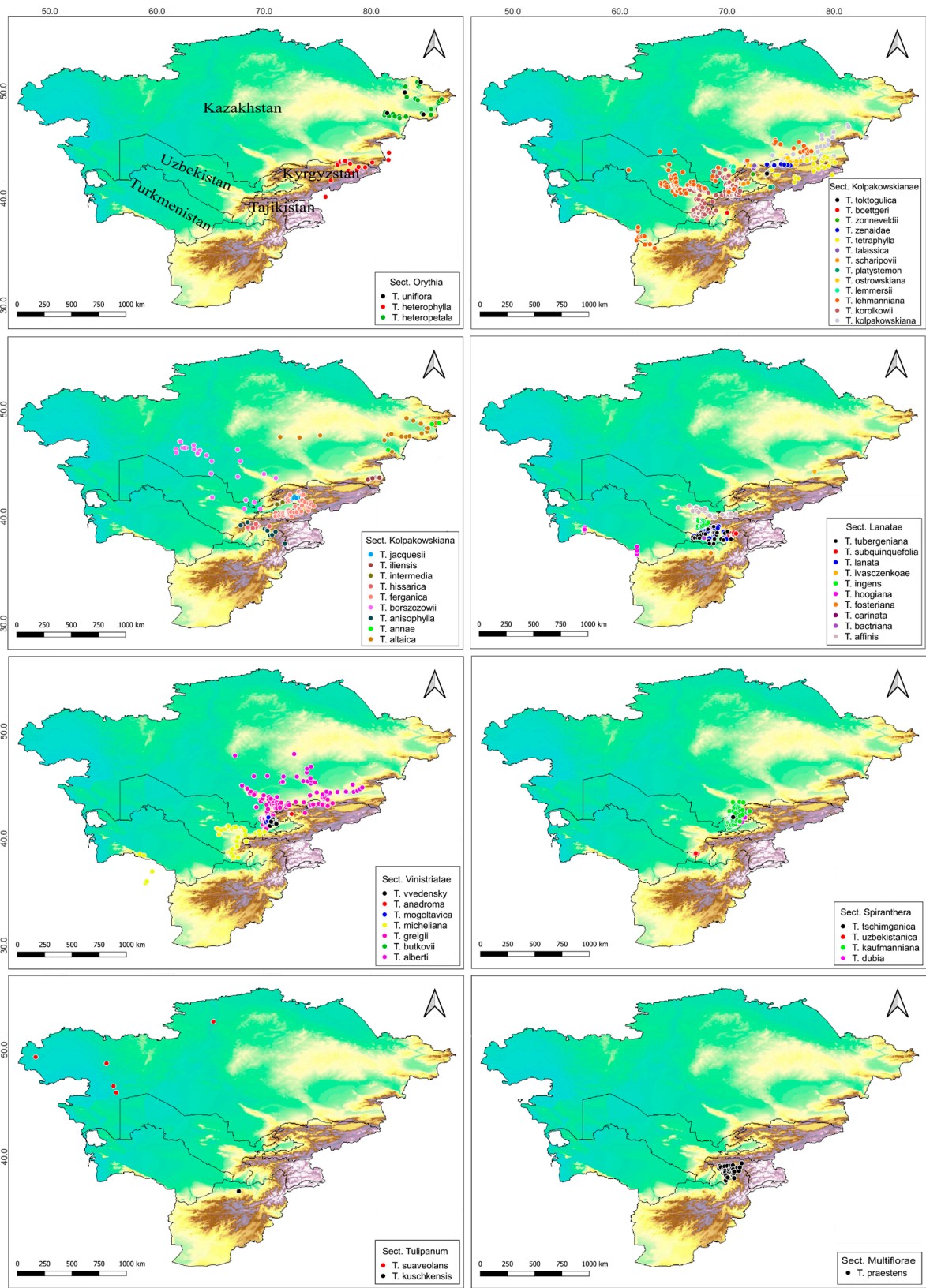

**Figure 6.** *Cont.*

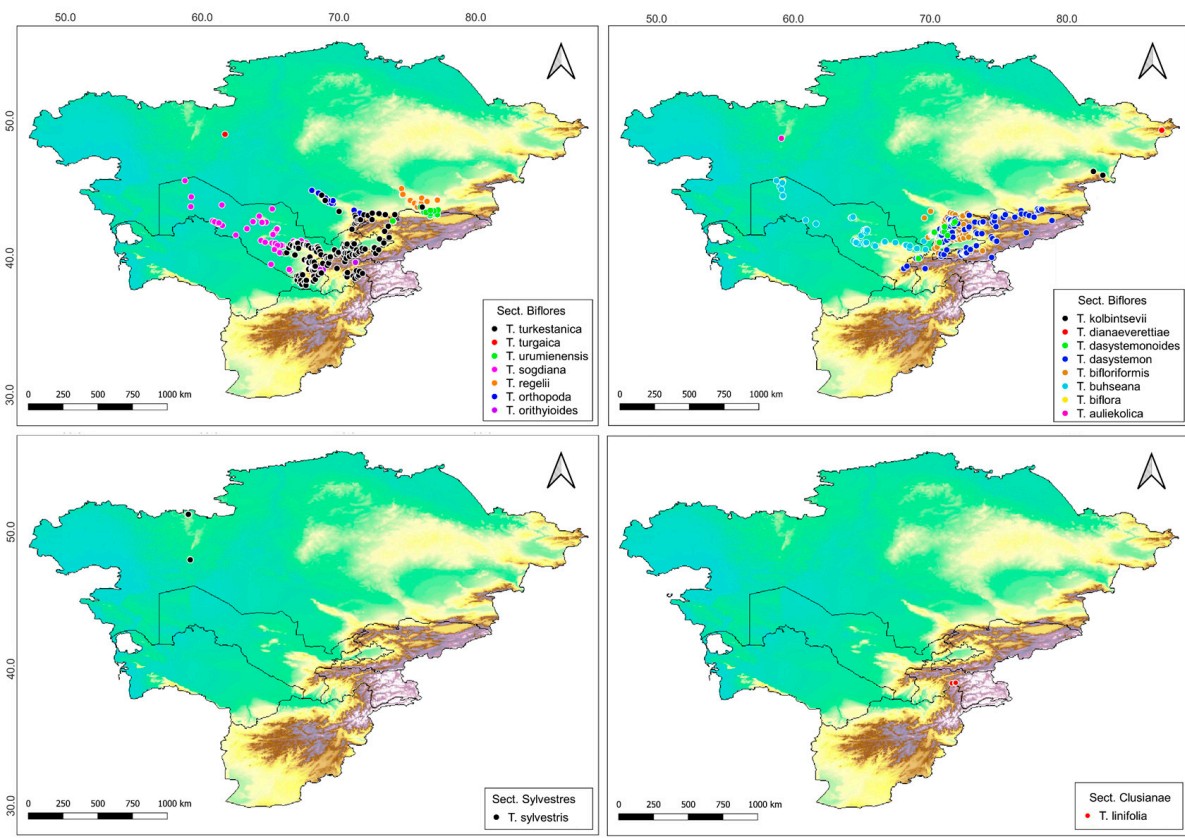

**Figure 6.** Distribution of tulips in relation to sections.

Two sects. *Biflores* and *Sylvestres* from subg. *Eriostemones* had 15 and 1 species, respectively, while sect. *Clusianae* from subg. *Clusianae* had a single species in Central Asia. Notably, almost all sections were in the mountainous regions of southeastern Central Asia.

## 4. Discussion

### 4.1. Species Range and Its Importance in Conservation

Botschantzeva [8] identified four sub-areas of diversity within the primary center of diversity (Central Asia) for *Tulipa.* The areas included the Turan plains, Pamir-Alay Mountains, Western Tian Shan Mountains, and Central Asian highlands. The findings of the current study align with Botschantzeva's observations. Adaptive potential is determined by genetic variation, that helps a population's traits to change in response to changes in their habitat [38], and genetic diversity in turn leads to greater adaptability in populations of a species. Interestingly, almost two-thirds of 66 species of *Tulipa* in Central Asia are in more than one ecoregion. This may highlight the fact that species with a wider ecoregional range (e.g., species of Groups 1, 2, and 3) are apparently quite adaptable (have a higher genetic diversity within a species population) and are at less risk of extinction if ecological fluctuations, global warming and climate change occur. Certain cases provide compelling evidence that rarity can be indicative of heightened vulnerability or conservation risk for plants in the face of climate change [39,40]. Notwithstanding, 24 species (Group 1) among 64 species of *Tulipa* occurred in only a single ecoregion, either the same or different, which may indicate that they are constrained by ecological barriers. However, there may be other explanations why these species (Group 1) have a restricted range. Some studies have provided evidence that plant species typically initiate their existence as smaller entities, emerging as "budding" offshoots from parental species, often within the same geographic range, and subsequently undergo expansion through niche evolution and/or wider dispersal over time [41,42]. Thus, phylogenetic and evolutionary studies can be

conducted on these species; however, this never means that these species should not be targeted under conservation as they have a limited range.

### 4.2. Ecoregional Distribution of Tulips in Central Asia

Ecoregions play a crucial role in the conservation of plant species by providing a framework for identifying and prioritizing areas of high biodiversity value [43]. By studying the dispersion of plant species within ecoregions, researchers can gain insights into the factors influencing species distribution, such as climate, soil, topography, and human activities [19]. This information is crucial for identifying areas of high conservation priority, predicting the potential impacts of environmental changes, and developing effective strategies for biodiversity conservation. In our study, the Gissaro-Alai open woodlands (GAOW) ecoregion was found to host the greatest number (almost two-thirds of 66 species) of the species of *Tulipa*. Most of this ecoregion nearly corresponds to the western Tian Shan (ca. 3440 vascular plants species and subspecies) and Pamir Alai mountains (ca. 4500–5000 vascular plants) [27,44,45]. We identified 14 species of *Tulipa* in this ecoregion that do not grow in neighboring ecoregions.

Previous studies highlighted Central Asia, especially the Pamir Alai and Tian Shan mountains (e.g., Ferghana Valley) as the center of biodiversity [8,46–48] and that it should be considered a priority area for conservation. Furthermore, the Tian Shan foothills and arid steppe (TSFAS) and Alai-Western Tian Shan steppe (AWTSS) also contained a relatively high number of tulip species, giving the impression that these areas may also contain high species diversity and a relative abundance of those species. It may also indicate that these areas might have had a unique geological history in comparison with the remaining ecoregions. However, in Central Asia, not all ecoregions are as suitable for tulips as others are. The AAMT, AMFFS, ASSD, CLD, CARW, EVS, JBSD, KSD, KS, and KDWFS ecoregions exhibit low suitability, or serve as habitats for only a limited number, ranging from one to four species of *Tulipa*. The reasons are that these ecoregions have unfavorable conditions not only for tulips but for whole plant diversity. For example, the AAMT ecoregion is mostly considered as an alpine zone and consists of higher altitudes, a low temperature and widespread permafrost soil [49]. The next ecoregions are described as dry and cold, as having an extreme variability in temperature between winter and summer, as well as a richness in forest, plains and deserts, and territories [50–53]. Our findings underscore the importance of careful investigation and evaluation in understanding the complex relationships between plant species and their ecological context. Accordingly, in our study, analysis of the ecoregional distribution of species of *Tulipa* in Central Asia revealed that species richness is concentrated in specific regions, with Pamir-Alai and Western Tian Shan identified as hotspots for tulips. Most parts of the Western Tian Shan and Pamir Alai GAOW ecoregion host unique tulips (or tulips endemic to this area). We strongly recommend that this ecoregion be considered as a leading priority in conservation plans and management.

The relationship between altitude and species richness for tulips in Uzbekistan revealed a stronger correlation. The 700–2200 m a.s.l. range was the most suitable for tulip diversity and within this range the ancestral lineages started to appear [14]. Our study area (about nine times larger) analysis is also nearly consistent with this finding, indicating that elevation is the most important indicator in the evolutionary dispersal of tulips. Elevational species richness is given in Table S3.

### 4.3. Phytogeographical Distribution

The goal of phytogeographical studies is to understand the reasons for the range of species, which can be influenced by factors such as their origin, dispersal and evolution [54]. According to Kamelin [27] the Afghano-Turkestan province is a hotspot for plants, hosting ~5500 species with approximately 1800 endemics and especially the Ferghana Valley districts have a special interest in it. In previous literature [8,55], the Pamir Alai and Western Tian-Shan mountains which cover most of the Ferghana Valley phytogeographic district

were considered as a primary biodiversity center for tulips. In addition, we identified this district as the leading biodiversity hotspot for tulips. If we take phytogeographical studies goal into account, we may consider that most tulips in Middle Asia originated in and diversified from the Afghano-Turkestan province area, especially from the Ferghana Valley phytogeographic district (25 tulipa species). From here, most tulips dispersed toward neighboring districts. For example, Mogoltau-Kurama (13 species) and Kukhistan (11 species) are adjacent to the Ferghana Valley phytogeographic district. Our preliminary idea can be only checked by molecular data (ancestral area reconstruction). We suppose elevational and climatic factors are favorable for species of *Tulipa* in these areas. However, the presence of species of *Tulipa* in multiple phytogeographic districts suggests that some of them have adapted to different environmental conditions, allowing them to survive in diverse habitats. This adaptability may have contributed to their success in colonizing new areas and expanding their range. The findings from this analysis provide insights into the diversity and distribution of species of *Tulipa* in Soviet Middle Asia, highlighting the importance of adaptation and survival strategies in the face of changing environmental conditions. This information can be useful in conservation efforts, as it highlights areas that may require more attention to protect and preserve the diversity of tulips.

*4.4. Sectional Distribution*

According to the World Flora Online database [56], 107 species of *Tulipa* are recognized. Among them, we determined that 66 species (61%) occur in Central Asia. The most recently updated infrageneric classification of *Tulipa* was by Veldkamp and Zonneveld [6]. According to their classification, subg. *Tulipa* has 7 sections. Interestingly, in our study we determined 6 sections of the subgenus *Tulipa* in Central Asia that have the highest number of species, 22 in sect. *Kolpakowskianae*, and the lowest number of species, 1 in sect. *Multiflorae*. So far, four species are in the subg. *Clusianae* sect. *Clusianae* [55] and only one species (*T. linifolia*) is in Central Asia. Similarly, four species were assigned to the subg. *Orythia* sect. *Orythia* [55]. Surprisingly, we determined that three of those species (*T. heteropetala, T. heterophylla* and *T. uniflora*) occur in Central Asia. Worldwide, within the subg. *Eriostemones* three sections, *Biflores*, *Saxatiles* and *Sylvestres*, were recognized [6,55]. The section *Biflores* has around 20 species [6,55,57,58]; in this study, we determined that 15 of those species are in the study area. We found only a single species (*T. sylvestris*) of sect. *Sylvestres* in our study area. Concluding the sectional distribution of tulips in Central Asia, we can say that sects. *Kolpakowskianae* and *Biflores* have the highest number of species and are specific to the Central Asian flora.

**5. Conclusions**

Central Asia, often regarded as a "nature imperiled" region, demands rigorous conservation efforts [3]. Our comprehensive analysis of *Tulipa* species distribution patterns sheds light on critical areas for conservation.

Hotspots: The Gissaro-Alay open woodlands ecoregion, harboring 41 species, emerges as a leading hotspot. Additionally, the Tian-Shan foothill arid steppe and the Alay-Western Tian-Shan steppe are significant *Tulipa* habitats. A species richness map within the botanical-geographical districts of Soviet Middle Asia highlights the Ferghana valley, boasting 25 *Tulipa* species. The Mogoltau-Kurama, Kukhistan, Gissar, and Kugitang-Baisun districts exhibit moderate species diversity.

Widespread Species: Biogeographic analyses reveal that *T. altaica* and *T. turkestanica* dominate across ecoregions and phytogeographic districts.

Endemism: Within individual ecoregions and phytogeographic districts, we observe an endemism rate. Approximately one-third of the species in both ecoregions and phytogeographic districts are endemic.

Historical Influences: The concentration of species richness in specific districts suggests that historical factors such as climate change, geological events, and biotic interactions have significantly impacted *Tulipa* distribution patterns.

In summary, our study provides valuable insights into the biogeography and conservation of *Tulipa* in Central Asia and Middle Soviet Asia. Future research should delve deeper into the historical processes that have shaped the distribution of *Tulipa* species across this remarkable region.

**Supplementary Materials:** The following supporting information can be downloaded at: https://www.mdpi.com/article/10.3390/d15121195/s1. Table S1: Ecoregional species richness; Table S2: Phytogeographical species richness; Table S3: Elevational species richness.

**Author Contributions:** Conceptualization, writing—original draft preparation, Methodology, T.A.; writing—review and editing, D.D., Z.Y., U.T. and L.C.; Supervision, K.T.; Supervision and Funding, W.S. All authors have read and agreed to the published version of the manuscript.

**Funding:** This work was supported by the State Projects "Tree of life: Monocots of Uzbekistan (PFI-5)" and "Grid mapping of the flora of mountainous regions of southern Uzbekistan" funded by the Institute of Botany, Academy of Sciences of the Republic of Uzbekistan and Science and Technology Basic Resources Investigation Program of China "Survey and Germplasm Conservation of Plant Species with Extremely Small Populations in Southwest China" (2017FY100100).

**Institutional Review Board Statement:** Not applicable.

**Data Availability Statement:** Data are contained within the article and supplementary materials.

**Acknowledgments:** We thank David Boufford of Harvard University for his kind help with the English editing.

**Conflicts of Interest:** The authors declare no conflict of interest.

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
