# Peer review of "Ecoregional and Phytogeographical Insights into the Distribution of Tulipa in the ‘Nature Imperiled’ Area of Central Asia for Effective Conservation"

_diversity, doi:10.3390/d15121195_

Round 1
Reviewer 1 Report
Comments and Suggestions for Authors
A nice detailed paper about a well-known and valued genus for the region, which should be valuable for conservation both of Tulipa and for regional biodiversity in general.
Author Response
Dear Reviewer! Thanks for your view about our research!
Reviewer 2 Report
Comments and Suggestions for Authors
Please follow the revisions included in the amended PDF point by point

Author Response
Thanks for reviewing our MS. Please see the attachment.

Reviewer 3 Report
Comments and Suggestions for Authors
The authors present an interesting study of the Central Asian Tulipa L. and identified the open forests of Gissaro-Alai as the most important conservation hotspot with 41 species. The Ferghana valley is also ideal for 25 species. About 61% of tulip species are found in Central Asia, and some sections have the highest diversity in this region. These findings contribute to the conservation of tulips in the region.
Introduction: 41-42: Change "many endemic species found nowhere else in the world" because is a pleonasm.
44-46: Reference for The International Union for the Conservation of Nature (IUCN) Red List of Threatened Species (2021)
58-65: It is not necessary to mention all these authors. Bibliographical references are sufficient
79-82: Delete the 'Sky Islands' mention. I don't see any point. At least in the Introduction.
93-102: This paragraph should be rewritten briefly
The next section should be renamed: add Materials (Materials and Methods) because that's where it starts - with herbarium materials.
119: What are LE and MW?
The methods used should be described in more detail.
Results:
135: It is not clear hoe authors created a map of species richness of Tulipa based on ecoregions
137: 'Gissaro-Alai open forests have been identified as the most important hotspot' - list the results first and then name the hotspot.
In fact it is not very clear how all the results were obtained.
It is obvious that this section needs to be rewritten according to the requirements of the journal.
Discussion mentions only a few bibliographical references.
There are many abbreviations, of ecoregions I understand, GAOW, TSFAS, AWTS and so on. Some are explained but most are not. However, I don't understand the point of them. They more confuse the reader.
Conclusions: unfortunately this section also needs to be rewritten because it does not follow the required rules. The same 'results' are repeated.
In summary, although the topic chosen for this study is very welcome the manuscript is not well written. There are gaps in all sections.
Author Response

(The authors gave the same response as above.)

Round 2
Reviewer 3 Report
Comments and Suggestions for Authors
Unfortunately, the authors did not take into account all the requirements. The manuscript was only partially improved. But there are a few paragraphs where it looks worse than before.
The introduction needs to be rewritten, at least in the second part. I ask the authors to read the journal requirements very carefully.
Are lines 99-100 an objective? then it should be moved to objectives.
102-105: This is not a textbook or a popularization journal.
Material and Methods
229: How (on what basis) was the habitat recorded for each specimen collected?
239: change the name of the region according to the current name
240: a bibliographic reference for ArcGIS v10.8
241: "Phytogeographic map of Soviet Middle Asia created by Kamelin [22]": This phrase is first found in the introduction and then in Material and Methods. Summarize this sentence.
The results are not convincing. The data are not presented and the analyses mentioned cannot be repeated.
As mentioned above, the discussion contains too few bibliographical references. It is not necessary to refer to Tulipa. Other common features can be found.
Conclusions: although rewritten, are not well written this time either.
Author Response
Thanks for your time to review our MS. Please see attached file with responses.
We believe that the changes that done by your recommendations will help to improve our manuscript. We appreciate your time and valuable suggestions!

Round 3
Reviewer 3 Report
Comments and Suggestions for Authors
The manuscript under review is in its third round. It has obviously been much improved and most of the reviewer's comments have been answered.
However, unfortunately, I have to make new comments:
143: In academic or scientific writing, especially if discussing methodologies, analyses, or specific tools (such as ArcGIS) that significantly impact your research or results, it's generally good practice to cite the software version used.
326 add a bibliographic reference
326, 328 avoid repetitions
While I appreciate your dedication to honesty in scientific inquiry and your willingness to provide the data to the Editor, the primary concern raised in my initial comment was the lack of persuasiveness in the reported results and the absence of presenting data that would enable the replication of your analyses. Your willingness to present the data to the Editor and your commitment to publishing it alongside the manuscript are commendable steps toward transparency. However, for a more comprehensive evaluation during the peer-review process, it would be beneficial to have access to the data, facilitating a thorough assessment of the study's findings.
Author Response
Dear Reviewer It's my pleasure to thank you as your comments and suggestions have contributed to the improvement of our MS. Please, find the attached file with responses. Thanks again!
